# Five-Year Cardiovascular Outcomes after Infective Endocarditis in Patients with versus without Drug Use History

**DOI:** 10.3390/jpm12101562

**Published:** 2022-09-22

**Authors:** Brandon Muncan, Alan Abboud, Aikaterini Papamanoli, Mark Jacobs, Allison J. McLarty, Hal A. Skopicki, Andreas P. Kalogeropoulos

**Affiliations:** 1Renaissance School of Medicine, Stony Brook University, Stony Brook, NY 11794, USA; 2Department of Medicine, Zucker School of Medicine, Northwell Health at Mather Hospital, Port Jefferson, NY 11777, USA; 3Division of Cardiology, Department of Medicine, Stony Brook University, Stony Brook, NY 11794, USA; 4Division of Cardiothoracic Surgery, Department of Surgery, Stony Brook University, Stony Brook, NY 11794, USA

**Keywords:** infective endocarditis, people who use drugs, cardiovascular events, cerebrovascular events, outcomes, mortality

## Abstract

Background: Disparities in treatment and outcomes of infective endocarditis (IE) between people who use drugs (PWUD) and non-PWUD have been reported, but long-term data on cardiovascular and cerebrovascular outcomes are limited. We aim to compare 5-year rates of mortality, cardiovascular and cerebrovascular events after IE between PWUD and non-PWUD. Methods: Using data from the TriNetX Research Network, we examined 5-year cumulative incidence of mortality, myocardial infarction, heart failure, atrial fibrillation/flutter, ventricular tachyarrhythmias, ischemic stroke, and intracranial hemorrhage in 7132 PWUD and 7132 propensity score-matched non-PWUD patients after a first episode of IE. We used the Kaplan–Meier estimate for incidence and Cox proportional hazards models to estimate relative risk. Results: Matched PWUD were 41 ± 12 years old; 52.2% men; 70.4% White, 19.8% Black, and 8.0% Hispanic. PWUD had higher mortality vs. non-PWUD after 1 year (1–3 year: 9.2% vs. 7.5%, *p* = 0.032; and 3–5-year: 7.3% vs. 5.1%, *p* = 0.020), which was largely driven by higher mortality among female patients. PWUD also had higher rates of myocardial infarction (10.0% vs. 7.0%, *p* < 0.001), heart failure (19.3% vs. 15.2%, *p* = 0.002), ischemic stroke (8.3% vs. 6.3%, *p* = 0.001), and intracranial hemorrhage (4.1% vs. 2.8%, *p* = 0.009) compared to non-PWUD. Among surgically treated PWUD, interventions on the tricuspid valve were more common; however, rates of all outcomes were comparable to non-PWUD. Conclusions: PWUD had higher 5-year incidence of cardiovascular and cerebrovascular events after IE compared to non-PWUD patients. Prospective investigation into the causes of these disparities and potential harm reduction efforts are needed.

## 1. Introduction

In the United States, the incidence of infective endocarditis (IE) among people who use drugs (PWUD) has nearly doubled over the last two decades [1]. In the context of the continuing opioid crisis, IE in PWUD has been attributed to nonsterile injection practices, reuse of contaminated injection equipment, inadvertent injection of particulate matter [2], and is a recognized risk of nonmedical drug use by methods other than injection as well [3]. Among PWUD, IE often occurs in younger patients with few preexisting conditions; however, these patients are frequently of lower socio-economic status who are often burdened with concomitant poverty, housing instability, and limited access to healthcare [1,4,5]. As rates of injection drug use have risen, the incidence of related infections has also significantly increased. In the United States, nonsterile drug injection has directly contributed to the synergistic epidemics of opioid overdose, human immunodeficiency virus (HIV), and hepatitis C [5,6].

While data suggest that IE in PWUD is associated with higher long-term mortality up to 10 years after the initial IE event [7,8], data on mortality are heterogenous. Compared to non-PWUD patients with native valve IE unrelated to drug use, observational studies report that survival for PWUD within 30 days of initial hospitalization ranges from marginally higher [9] to no mortality differences [10,11] to lower [7]. While IE as a complication of drug use has recently been framed as yet another developing epidemic secondary to drug use [12], outside of mortality data, other health outcomes of PWUD treated for IE are still largely unknown beyond index hospitalization. Moreover, many studies focus only on postoperative outcomes and complications in PWUD treated surgically [13]. Although past studies have reported an increased risk of ischemic stroke, myocardial infarction, heart failure, and ventricular arrythmia in survivors of IE compared to matched controls without IE [14], the literature remains sparse regarding morbidity between PWUD and non-PWUD.

In this study we aimed to describe 5-year mortality and rates of cardiovascular events (including myocardial infarction, heart failure, atrial fibrillation/flutter, and ventricular tachycardia/ventricular fibrillation [VT/VF]) and cerebrovascular (ischemic stroke and intracranial hemorrhage) in PWUD versus non-PWUD after a first episode of IE. In secondary analyses, the effect of medical vs. surgical treatment, sex and race were examined as well.

## 2. Materials and Methods

### 2.1. Study Design and Data Source

In this retrospective cohort study, data were sourced from the TriNetX Analytics Research Network, a global electronic health records network for clinical research purposes, comprising of data from over 80 million patients and 57 healthcare organizations (including academic and community hospitals, and specialty physician networks) at the time of this study, predominantly located in the United States. TriNetX, LLC, is a private initiative that partners with healthcare organizations to combine and harmonize real-world clinical data for research purposes (i.e., not for administrative or business purposes), with the goal of facilitating clinical trial design and conduct and enabling generation of real-world observational evidence. The harmonized data include demographics, standard measurements (including vital signs, lab results, medications, and genomic data), and diagnostic and procedural information recorded using standard coding systems, e.g., International Classification of Diseases, Tenth Revision (ICD-10) and Current Procedural Terminology (CPT) codes. The TriNetX Network is compliant with the Health Insurance Portability and Accountability Act (HIPAA), a federal law protecting the confidentiality of health information. All patient-level and aggregate data from TriNetX are fully de-identified for research purposes [1,2]. Importantly, the TriNetX platform maintains internal validity by monitoring real-time temporal trends in data [3]. External validation studies have confirmed the utility and reliability of TriNetX as a large-scale data source for retrospective research [4,5]. Data from a final search run on 28 October 2021, were used in this analysis. We followed the REporting of studies Conducted using Observational Routinely collected Data (RECORD) guidelines as a framework for this study [1,6].

### 2.2. Cohorts and Sample Selection

We queried the TriNetX Research Network database from 1 January 2010–31 December 2020, for patients with first-episode infective endocarditis (IE) with and without history of drug use according to the methodology used in a previous pivotal retrospective cohort study [7]. Inclusion criteria for IE were the ICD-10 codes I33, I38, and I39. Inclusion criteria for IE were (1) ICD-10 codes I33, I38, and I39; (2) age 16 to 64 at the time of the index IE event; (3) for PWUD designation, ICD-10 codes: F11, F13, F14, F15, F16, F18, F19, T40, and O99.32. Patients with a previous episode of IE were excluded.

To search for the majority of PWUD with IE while simultaneously excluding elderly patients who have higher baseline risk of IE due to age; we followed the validated methods of Cooper et al. in restricting patient age to 16–64 [8]. This methodology has been found to maximize specificity, despite differing from traditional age cutoffs in the literature [8]. In the ICD-10 coding system, there are no specific codes for IE related to drug use, nor are there specific codes for injection drug use. Therefore, we drew on published and validated methods to create a sensitive list of drug use codes which were used to capture PWUD: ICD-10 codes: F11, F13, F14, F15, F16, F18, F19, T40, and O99.32 [7,8,9,10]. Importantly, there are no specific codes to distinguish IE by common micro-organisms (i.e., separate codes for staphylococcal IE versus streptococcal IE, etc.) as these are contained in the ICD-10 codes I33, I38, and 139 [11,12]. 

We thus obtained two primary cohorts of interest: PWUD with IE and non-PWUD with IE.

### 2.3. Outcomes

We compared mortality and major cardiovascular and cerebrovascular morbidity at 5 years after an initial IE episode among PWUD and non-PWUD. Specifically, we examined all-cause mortality, ventricular tachycardia and/or ventricular fibrillation (VT/VF; ICD-10 codes I47.0, I47.2, I49.0), myocardial infarction (I21), heart failure (I50), atrial fibrillation and/or flutter (I48), ischemic stroke (I63), and intracranial hemorrhage (I61, I62). For each morbidity endpoint, we excluded patients with prior history of the endpoint of interest (i.e., before the index IE episode) from our primary and secondary analyses, thereby assessing only for new-onset or newly diagnosed conditions. Of note, removing patients with a prevalent condition/event of interest from matched cohorts does not introduce bias, as the proportion of removed patients is by definition similar between matched cohorts.

### 2.4. Statistical Analysis

Categorical and ordinal variables are expressed as a frequency and percentage of the total cohort, and continuous characteristics are expressed as mean ± standard deviation. Baseline characteristics were compared with standardized mean difference (SMD) between cohorts. We used the standardized mean difference (SMD) as the preferred method to assess balance between cohorts as statistical hypothesis testing methods are not appropriate when balancing cohorts with propensity score matching; the SMD is calculated as the difference in means divided by the estimate of the within-group standard deviation [13]. Matching was considered adequate if SMD was <0.1 [14]. Patients meeting inclusion criteria were 1:1 propensity score-matched using logistic regression for 54 variables including demographic characteristics, comorbidities, surgical history, and psychosocial characteristics, all before first episode IE which are summarized in Table 1. Matching was performed in the TriNetX Live platform (TriNetX, Cambridge, MA, USA) which uses nearest neighbor matching with a caliper of 0.1 standard deviation. Cumulative 5-year incidence of the outcomes of interest after the index IE event was calculated with the Kaplan–Meier method and compared with the log-rank test between cohorts. Of note, patients with the same event or condition of interest present at baseline were excluded in secondary analyses for the corresponding outcome, to query only for new-onset events as described above. We generated corresponding hazard ratios using Cox proportional hazards models. The proportionality of hazards was examined with the Schoenfeld residuals. All outcomes met the proportional hazards assumption except for all-cause mortality. We therefore calculated hazard ratios for mortality for 3 separate periods: (1) <1 year; (2) 1–3 years; (3) 3–5 years after the index IE event; the proportional hazards assumption was valid within these periods.

Differences in post-IE outcomes by sex and race/ethnicity have been noted in the literature [15,16,17]. Therefore, we performed secondary analyses of outcomes in male PWUD vs. non-PWUD, female PWUD vs. non-PWUD, and among PWUD vs. non-PWUD by race. Additionally, we performed secondary analysis of IE treated medically versus surgically as differences in short-term outcomes were also cited in previous studies [18,19]. We used the same inclusion and exclusion criteria as our main analysis, with independent propensity score matching for each comparison. Hazard ratios between sex and race subgroups for the outcomes of interest were compared using the asymptotic method [20]. All statistical tests were performed in the TriNetX Live platform (TriNetX, Cambridge MA), which uses R version 3.2–12 and SAS version 9.4, except for comparison of hazard ratios between sex and race subgroups, which were calculated directly. Statistical significance for all tests was set at a two-tailed *p* value < 0.05 unless otherwise noted.

## 3. Results

### 3.1. Baseline Characteristics

Before propensity score matching, our sample consisted of 15,573 PWUD and 49,521 non-PWUD across 54 healthcare organizations in the United States. Geographically, 41% of the PWUD cohort was from the South, followed by 34% from the Northeast, 15% from the Midwest, 9% from the West, and 1% from unknown or undocumented region. Similarly, 50% of the non-PWUD cohort was from the South, followed by 26% from the Northeast, 15% from the Midwest, 7% from the Western United States, and 2% did not have a documented geographic region. The PWUD cohort was younger (age 39.2 ± 11.2 vs. 45.8 ± 12.8 years; SMD: 0.479), more likely to identify as White or Caucasian (75.3% vs. 67%; SMD, 0.184), and more likely to have comorbid acute and/or subacute cardiac, respiratory, renal, hematologic, infectious, and psychiatric conditions. Of note, endocrine and metabolic comorbidities including diabetes mellitus and obesity were more common among non-PWUD. After 1:1 propensity score matching for 54 characteristics, a total of 7,132 PWUD were compared with 7,132 non-PWUD; SMD for all matched characteristics was <0.1, indicating adequate balance between cohorts. Baseline characteristics before and after matching are shown in Table 1. Outcomes before matching are presented in Appendix A.

### 3.2. Five-Year Mortality

After matching, cumulative 5-year mortality was 25.7% and 23.0% in PWUD and non-PWUD, respectively. Mortality was highest in both cohorts in the first year after IE. However, mortality risk was not proportional between groups throughout the 5-year follow up. Non-PWUD had marginally higher mortality (12.2%) compared to PWUD (11.6%) during the first year, but the difference was not significant (log-rank *p* = 0.303). After 1 year, a reversal in mortality trend was noted: PWUD had greater mortality (9.2%) compared to non-PWUD (7.5%) in the isolated 1–3-year period (*p* = 0.032) and in the isolated 3–5-year period (7.3% vs. 5.1%; *p* = 0.020); see Figure 1 and Table 2.

### 3.3. Five-Year Cardiovascular Events

Outside of mortality, all other outcomes of interest demonstrated proportional hazards. Compared to non-PWUD, PWUD had significantly higher rates of myocardial infarction, heart failure, ischemic stroke, and intracranial hemorrhage at five years post-IE event; VT/VF was comparable between groups, as was atrial fibrillation/flutter. See Figure 2 and Table 2.

### 3.4. Secondary Analyses

#### 3.4.1. Surgical and Medical Treatment

Of the total 15,573 PWUD and 49,521 non-PWUD before matching, 1303 (8.4%) and 2099 (4.2%), respectively, were treated surgically (chi-square *p* < 0.001). Most surgically treated PWUD (53%) had interventions on the tricuspid valve while most non-PWUD (51%) had interventions on the aortic valve. After propensity score matching, we compared the outcomes of 392 surgically treated PWUD vs. 392 non-PWUD, Appendix A. Incidence and risk of all-cause mortality, VT/VF, myocardial infarction, heart failure, atrial fibrillation/flutter, ischemic stroke, and intracranial hemorrhage were comparable between surgical groups.

Among medically treated patients, 6390 PWUD and 6390 non-PWUD were compared after matching, Appendix A. Compared to non-PWUD, PWUD had higher rates of myocardial infarction (9.2% vs. 6.2%; *p* = 0.001), and intracranial hemorrhage (4.0% vs. 2.4%; *p* = 0.015). Incidence of other outcomes was not significantly different between cohorts.

#### 3.4.2. Sex

Among men, 10% of PWUD were treated surgically vs. 7% of non-PWUD (chi-square *p* < 0.001). PWUD had a higher rates of heart failure (20.3% vs. 17.7%; *p* = 0.007) and intracranial hemorrhage (5.2% vs. 3.6%; *p* = 0.044), but rates of all other outcomes of interest were comparable vs. non-PWUD, Appendix A. Among women, 10.0% of PWUD were treated surgically vs. 5.0% of non-PWUD (chi-square *p* < 0.001). PWUD had significantly higher rates of all-cause mortality (24.8% vs. 20.9%; *p* = 0.019), myocardial infarction (9.1% vs. 6.0%; *p* = 0.002), heart failure (19.2% vs. 15.1%; *p* = 0.010), and ischemic stroke (8.3% vs. 5.2%; *p* < 0.001) vs, non-PWUD, Appendix A. Rates of VF/VT, atrial fibrillation/flutter, and intracranial hemorrhage were comparable between groups. Stroke among PWUD was significantly more pronounced in women (*p* = 0.026 for the difference of HRs).

Importantly, the elevated risk of mortality was most apparent in female PWUD, as male PWUD did not have elevated mortality risk; nevertheless, the risk difference did not reach statistical significance (*p* = 0.062 for the difference of HRs). 

#### 3.4.3. Race

Among White/Caucasian patients, 11% of PWUD were treated surgically vs. 7% of non-PWUD (chi-square *p* < 0.001); 7% vs. 5%, respectively among Black/African American patients (chi-square *p* < 0.001), and 10% vs. 4%, respectively among patients of other races (chi-square *p* < 0.001). After matching, among patients identifying as White/Caucasian, PWUD had significantly higher incidence of VT/VF (6.4% vs. 4.9%; *p* = 0.030), myocardial infarction (8.3% vs. 5.7%; *p* = 0.002), heart failure (19.4% vs. 12.9%; *p* < 0.001), ischemic stroke (7.5% vs. 5.8%; *p* = 0.022), and intracranial hemorrhage (4.2% vs. 2.6%; *p* = 0.009) compared to non-PWUD, Appendix A. Among patients identifying as Black/African American, rate of ischemic stroke was higher among PWUD (12.6% vs. 8.5%; *p* = 0.001), and atrial fibrillation/flutter was higher among non-PWUD (11.8% vs. 14.3%; *p* = 0.046), Appendix A. Among patients identifying as other race, there were no instances of myocardial infarction or intracranial hemorrhage among PWUD and there were no instances of VF/VT or ischemic stroke among non-PWUD. Nevertheless, rate of ischemic stroke was higher among PWUD of other races (14.0% vs. 0.0%; *p* = 0.026); however, the sample size of patients identified as other races was small and thus these results need to be interpreted with caution.

The mortality risk associated with drug use was only evident in White/Caucasian patients, although the risk differential in White/Caucasian vs. Black/African American patients did not reach statistical significance (*p* = 0.059 for the difference of HRs). However, the elevated risk of heart failure among PWUD was present only in White patients and the difference in HR was statistically significant (*p* = 0.003 for the difference of HRs).

## 4. Discussion

In this propensity score-matched retrospective cohort study, we found higher 5-year risk of mortality, myocardial infarction, heart failure, ischemic stroke, and intracranial hemorrhage among PWUD with IE compared to their non-PWUD counterparts. Although death in the first year after the initial IE event was more common in non-PWUD, mortality was consistently and significantly higher in PWUD after the first year. Secondary analyses suggested that mortality risk among PWUD was higher in women. White PWUD had more pronounced risk for heart failure. Outcomes were comparable in PWUD vs. non-PWUD after surgical treatment of IE. Together, these findings indicate poorer outcomes of IE in PWUD compared to IE unrelated to drug use, despite matching for other patient characteristics and medical conditions.

Our findings on mortality are similar to those reported previously. Rates of immediate postoperative and early-term mortality have been shown to be either comparable [21] or lower [7] among PWUD, likely attributed to younger age, fewer comorbidities, and overall lower surgical risk compared to non-PWUD [22]. In contrast, mid- and long-term data consistently show higher mortality rates among PWUD, with etiologies ranging from recurrent IE with cardiac sequalae [22] to other drug-use related pathologies including fatal overdose [23]. Although we did not investigate causes of death, we did match for several psychosocial and drug use-related characteristics, likely minimizing confounding from external factors at baseline. Although unobserved residual confounding is likely, the higher rate of cardiac complications among PWUD suggests sequalae from the index event or synergistic effects of persistent substance abuse with IE sequelae.

Risk of myocardial infarction and heart failure was higher among PWUD in our study, adding to the available evidence base, as cardiac sequelae of IE in PWUD remain under-reported [24,25]. A landmark propensity score-matched retrospective study of 8,494 patients with IE reported higher risk of sudden cardiac death, myocardial infarction, and readmission for heart failure among IE patients compared to healthy controls over a mean follow-up time of 4.1 ± 3.1 years; although drug use was an independent risk factor for recurrent IE, longitudinal outcomes among PWUD versus non-PWUD were not considered [26]. High rates of heart failure in IE patients have been reported; however, data in PWUD population are sparse [25]. Furthermore, we observed higher rates of both ischemic stroke and intracranial hemorrhage among PWUD. This is important, as conventional teaching emphasizes tricuspid valve IE in PWUD with septic pulmonary emboli as a common and feared consequence [27,28]; nevertheless, left-sided IE has been reported to account for up to 35% of drug use related-IE, which increases the risk of embolic stroke [27].

In terms of potential etiologies for poorer outcomes in PWUD, myocardial infarction is a known risk of stimulant use (i.e., cocaine, amphetamines) via coronary vasospasm [29], and opioid use has been associated with an increased inflammatory response, oxidative stress, higher risk of dyslipidemia, and consequent coronary artery disease [30]. Although embolic myocardial infarction directly related to intracardiac infection is infrequent [30], continued drug use may contribute to a greater risk of vasospasm and atheroma formation, both of which are potential mechanisms of myocardial ischemia. Previous studies have acknowledged several potential mechanisms of both ischemic and hemorrhagic stroke after IE; however, much like cardiac consequences, robust data from PWUD are lacking. Cardioembolism for example, is recognized as a cause of stroke among patients with IE, particularly those with long-standing atrial fibrillation [29,31]. Of note, we observed that atrial fibrillation/flutter was more common in non-PWUD patients after IE, suggesting that the development of atrial dysrhythmias may not completely explain the disproportionate stroke risk in PWUD. Additionally, mechanisms including septic emboli from chronically infected valves, cerebral infections from continued unsterile drug injection, cerebral vasospasm, and hypertensive crises have been posited as potential explanations for increased cerebrovascular events in PWUD [32].

In our study, PWUD had higher rates of surgical treatment vs. non-PWUD. Five-year outcomes among PWUD undergoing surgery for IE did not differ significantly from those in non-PWUD. Our results agree with trends reported in the literature, although mortality among PWUD tended to be higher in our cohorts. Kim et al. reported a 5-year mortality of 21.1% in surgically treated PWUD, compared to 23.9% in non-PWUD (*p* = 0.39) [33]. After propensity score-matching, early risk was lower in PWUD (HR, 0.32; 95% CI, 0.08–1.36) but late risk was higher (HR, 2.07; 95% CI, 0.78–5.48) compared to non-PWUD, and risk of postoperative complications was higher in PWUD with IE [33]. A recent meta-analysis of 19 studies, reported a 5-year mortality of 19% in PWUD vs. 15% in non-PWUD among surgically treated IE patients; reoperation rates were higher among PWUD, but other outcomes were not reported [34]. The medically treated cohorts, i.e., the vast majority of patients, reflect the results in the total PWUD and non-PWUD populations. This emphasizes the problems with, e.g., incomplete antibiotic treatment courses and inadequate follow up among PWUD, and thus the need to tackle systemic issues, including services to treat substance use concomitantly with IE, overcoming preconceptions about surgical outcomes in eligible patients, and potentially the development of guidelines specific to IE among PWUD [35].

In our study, the mortality differential between PWUD and non-PWUD was driven by poorer outcomes among women, although this difference was marginally significant. Of note, the elevated stroke risk in PWUD was significantly higher in women. The reasons for these disparities are unclear. Although previous work reported that women with IE without drug use history are less likely to receive valve surgery despite meeting indications, and therefore had higher mortality [36], our results diverge somewhat from trends reported previously [7,37], as women were equally treated with surgery (10% for both sexes) in our study. In White patients, VF/VT, myocardial infarction, heart failure, ischemic stroke, and intracranial hemorrhage were higher in PWUD. Among Black patients, ischemic stroke was significantly higher among PWUD. Previous work has acknowledged the systemic and institutional disadvantages faced by women and racial/ethnic minorities when receiving care for IE and for substance use disorders in general [15,38]. The recommendations of previous authors and the American Association for Thoracic Surgery, including the need to connect PWUD to addiction medicine services, with special attention to vulnerable demographic populations, are especially relevant [18,39,40].

Our findings have several clinical implications. Most cases of IE in PWUD are due to non-sterile injection, and therefore education about, and provision of, safe injection material is paramount. Previous data have demonstrated that IE is up to six times more common in PWUD who did not practice safe injection [41], and a growing body of evidence suggests that outpatient harm reduction and addiction medicine services play a key role in prophylaxis and management of IE [42,43,44]. Harm reduction refers to strategies and techniques of reducing the negative health outcomes of drug use (i.e., overdose, infection, etc.) independent of drug use cessation; the goal is not total abstinence from drugs, but rather the promotion of safe use [45]. Harm reduction can be encouraged and practiced in healthcare and community settings and has a clear role in the prevention of drug-use related infections [41,46,47]. Where feasible, harm reduction services should be included in the multidisciplinary approach to IE [40,42,48].

## 5. Limitations

Our study has several limitations. First, despite the large sample size, there is a lack of mechanistic data on valvular and other structural sequelae of the index IE event, which may have helped explain our results. Second, although mitigated in part by the large number of contributing healthcare organizations to the TriNetX network and propensity score matching, the treatment algorithms for IE vary across institutions, which may have influenced outcomes, as stratification or clustering by organization is not possible in TriNetX. Third, the TriNetX database comprises of aggregate, not patient-level data. Therefore, the accuracy of the diagnoses and outcomes cannot be individually confirmed, and data-entry error and/or miscoding may in part confound the findings. On the other hand, the large sample size and the demonstrated internal and external validity of the TriNetX network should adequately balance this limitation. Fourth, due to the nature of ICD-10 coding, specific data on microorganisms implicated in IE cases was not available. Fifth, we acknowledge that our primary endpoint of all-cause mortality may not represent death directly as a result of IE or sequelae of IE. As postmortem data are unavailable, and linkage to death certificates would not be possible due to the deidentified nature of the TriNetX data, exact causes of death are impossible to ascertain; nevertheless, discrepancies in all-cause mortality after IE in PWUD still represents an important metric to investigate and follow. Sixth, the majority of data in both cohorts represented the South and Northeast regions of the United States. Although the geographic distribution was comparable between cohorts, limited availability of data from other regions may potentially bias results based on location, which we acknowledge and furthermore suggest for future topic of investigation. Lastly, we did not disaggregate outcomes of IE by drug type as polydrug use is common among PWUD and may play a synergistic role in pathogenesis and prognosis of disease.

## 6. Conclusions

Overall, the treatment of IE in PWUD remains complex and data regarding outcomes beyond the short-term are limited. We report higher 5-year rates of cardiovascular and cerebrovascular events in PWUD with IE compared to patients without drug use history, with potentially higher mortality rates among women who use drugs. Further research into mechanistic explanations for these outcomes is warranted. Guideline and practice changes should target minimizing the gap in IE care received by PWUD. Future work in drug-specific outcomes of IE will help understand the pathophysiology and to develop an evidence basis for treatment of IE in PWUD.

## Figures and Tables

**Figure 1 jpm-12-01562-f001:**
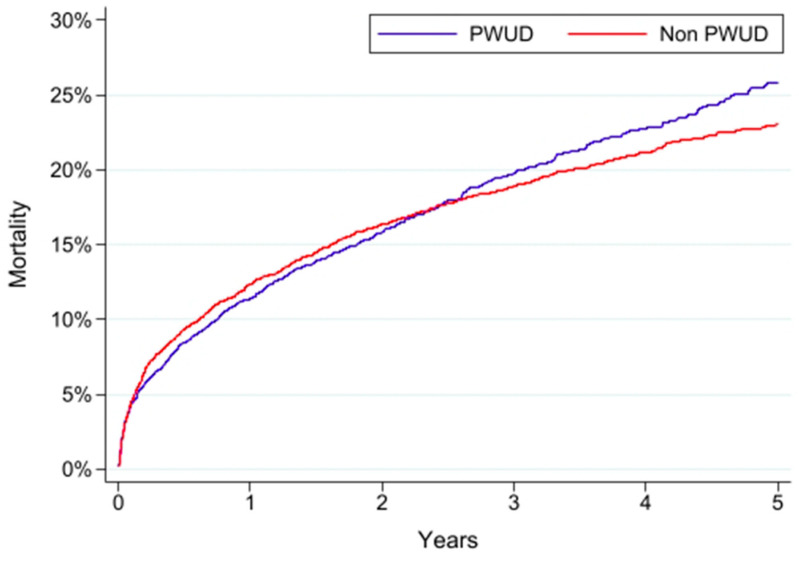
Five-year cumulative incidence of mortality after a first episode of infective endocarditis in people who use drugs (PWUD) vs. non-PWUD patients.

**Figure 2 jpm-12-01562-f002:**
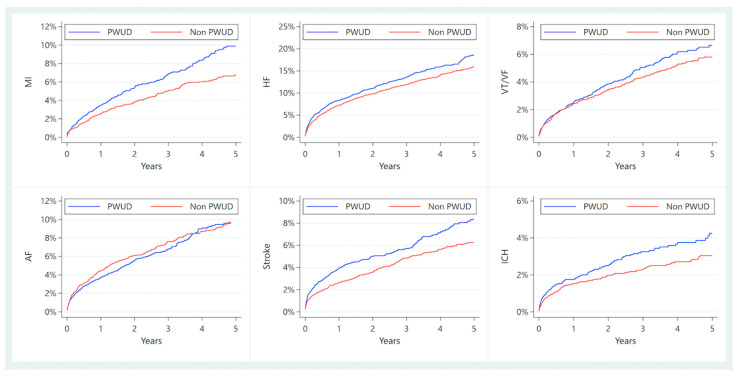
Five-year cumulative incidence of cardiovascular and cerebrovascular events after a first episode of infective endocarditis in people who use drugs (PWUD) vs. non-PWUD patients. We excluded patients who had recorded history of each outcome of interest at baseline, respectively (i.e., those with known AF were excluded from AF analysis) to only capture new-onset events. AF: atrial fibrillation or flutter; HF: heart failure; ICH: intracranial hemorrhage; MI: myocardial infarction; PWUD: persons who use drugs; VF: ventricular fibrillation; VT: ventricular tachycardia.

**Table 1 jpm-12-01562-t001:** Baseline characteristics at or before the time of infective endocarditis diagnosis before and after propensity score matching.

	Before Matching			After Matching		
Characteristic (ICD-10/CPT Code)	PWUD (N = 15,573)	Non-PWUD (N = 49,521)	SMD	PWUD (N = 7132)	Non-PWUD (N = 7132)	SMD
**Demographics**						
Age, years	39.2 ± 11.2	44.9 ± 12.8	0.476	41.3 ± 11.5	42.0 ± 13.2	0.055
Male, %	51.4	52.6	0.024	52.2	52.2	0.001
Race, %						
White	75.3	67.1	0.183	70.4	70.4	0.001
Black or African American	15.6	19.1	0.094	19.8	21.0	0.030
Hispanic or Latino, %	6.2	8.9	0.104	8.0	8.3	0.008
Body mass index, kg/m^2^	26.7 ± 7.1	29.5 ± 7.9	0.378	27.8 ± 7.6	28.0 ± 7.8	0.022
**Comorbidities, %**						
** *Cardiovascular and Related* **						
Essential hypertension (I10)	35.1	35.6	0.011	39.6	44.4	0.097
Diabetes mellitus (E08–E13)	17.6	18.4	0.020	21.3	24.2	0.069
Overweight, obesity (E65–E68)	15.5	17.7	0.059	19.0	20.9	0.047
Ischemic heart disease (I20–I25)	21.2	16.4	0.125	21.8	24.3	0.058
Previous cardiac surgery (02 procedure codes)	23.1	12.4	0.284	19.2	21.1	0.048
Prosthetic heart valve (Z95.2) *	7.5	6.4	0.044	5.9	6.8	0.039
Chronic rheumatic heart diseases (I05–I09)	17.2	9.8	0.218	12.1	13.4	0.039
Cerebrovascular disease (I60–I69)	13.3	8.3	0.161	11.7	13.5	0.055
Atrial fibrillation and flutter (I48)	8.1	9.4	0.044	9.1	10.4	0.046
Heart failure (I50)	19.1	15.5	0.097	19.7	22.0	0.056
Pulmonary heart disease (I26–I28)	23.2	8.7	0.402	15.3	16.6	0.035
Congenital cardiovascular malformations (Q20–Q28)	6.0	8.9	0.109	5.9	6.2	0.012
** *Respiratory* **						
Chronic lower respiratory diseases (J40–J47)	25.5	14.7	0.271	24.8	27.2	0.055
Influenza and pneumonia (J09–J18)	28.1	12.6	0.395	23.6	25.3	0.040
** *Renal* **						
Acute kidney failure and chronic kidney disease (N17-N19)	32.1	19.3	0.296	29.0	32.0	0.065
Other diseases of the urinary system (N30–N39)	21.6	12.1	0.258	20.6	22.2	0.039
Other disorders of kidney and ureter (N25–N29)	13.2	9.5	0.117	14.0	15.6	0.044
** *Infectious* **						
Human immunodeficiency virus (B20)	3.2	1.1	0.148	2.8	3.1	0.023
Viral hepatitis (B15–B19)	34.3	2.9	0.882	13.6	13.0	0.016
Skin and soft tissue infections (L00–L08)	37.3	11.9	0.617	30.1	32.4	0.049
Bacterial and viral infectious agents (B95–B97)	37.5	12.7	0.597	26.0	27.9	0.042
Mycoses (B35–B49)	15.8	8.1	0.237	14.3	16.0	0.048
** *Hematologic and Oncologic* **						
Aplastic anemia and bone marrow failure syndromes (D60–D64)	38.9	22.0	0.373	33.7	37.5	0.079
Coagulopathies (D65–D69)	21.1	10.5	0.292	17.5	19.6	0.054
Nutritional anemias (D50–D53)	15.4	8.5	0.215	14.4	15.7	0.038
Personal history of malignant neoplasm (Z85)	4.8	4.4	0.017	5.9	6.7	0.034
** *Mental Health* **						
Mood disorders (F30–F39)	41.6	13.6	0.661	32.7	35.5	0.058
Anxiety disorders (F40–F48)	41.1	14.2	0.631	31.9	33.6	0.037
Schizophrenia and non-mood psychotic disorders (F20–F29)	6.5	1.4	0.263	4.6	4.5	0.007
Poisoning/overdose (T36–T50)	28.4	7.0	0.584	21.3	22.6	0.030
Long term opiate use (Z79.891)	18.6	10.1	0.244	18.8	20.9	0.052
Alcohol related disorders (F10)	17.7	3.6	0.471	12.0	12.9	0.025
Cannabis related disorders (F12)	16.9	1.2	0.568	7.2	6.4	0.028
Nicotine dependence (F17)	57.6	11.8	1.096	39.1	41.0	0.039
** *Other* **						
Pregnancy, childbirth, and the puerperium (O00–O9A)	8.8	2.7	0.263	6.3	6.6	0.011
Health hazards due to socioeconomic circumstances (Z55–Z65)	13.4	2.2	0.425	7.3	7.2	0.005

Continuous variables are presented as mean ± standard deviation. PWUD with IE was defined using the ICD-10 codes F11, F13, F14, F15, F16, F18, F19, T40, and/or O99.32 at the same time or before the IE codes I33, I38, I39. * Prosthetic valves include those implanted by surgical and interventional approaches.

**Table 2 jpm-12-01562-t002:** Cumulative 5-year outcome incidence after propensity score matching.

	PWUD (N = 7132)	Non-PWUD (N = 7132)		
Outcome	Cumulative Incidence	Cumulative Incidence	HR (95% CI)	Log-Rank *p*
All-cause mortality				
Cumulative 5-year	25.7%	23.0%	1.04 (0.96–1.13)	0.363
<1 Year	11.6%	12.2%	0.95 (0.85–1.05)	0.303
1–3 Years	9.2%	7.5%	1.29 (1.02–1.42)	0.032
3–5 Years	7.3%	5.1%	1.39 (1.05–1.86)	0.020
VT/VF	6.6%	5.8%	1.13 (0.94–1.36)	0.211
Myocardial infarction	10.0%	7.0%	1.37 (1.16–1.62)	<0.001
Heart failure	19.3%	15.2%	1.22 (1.08–1.37)	0.002
Atrial fibrillation/flutter	10.1%	10.5%	0.90 (0.77–1.04)	0.154
Ischemic stroke	8.3%	6.3%	1.32 (1.12–1.57)	0.001
Intracranial hemorrhage	4.1%	2.8%	1.38 (1.08–1.76)	0.009

Incidence calculated with Kaplan–Meier estimator. Abbreviations. CI: confidence interval; HR: hazard ratio; PWUD: people who use drugs; VT/VF: ventricular tachycardia/ventricular fibrillation. We excluded patients who had recorded history of each outcome of interest at baseline, respectively (i.e., those with known atrial fibrillation were excluded from atrial fibrillation analysis).

## Data Availability

Data are available to Registered TriNetX Users.

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
