# Peer review of "Five-Year Cardiovascular Outcomes after Infective Endocarditis in Patients with versus without Drug Use History"

_jpm, 2022, doi:10.3390/jpm12101562_

Round 1

Reviewer 1 Report

The authors present an original article describing the five-year cardiovascular outcomes after IE in patients with and without drug use history.

Authors affirm, trough the results presented, that 5-year mortality, myocardial infarction, heart failure, ischemic stroke and intracranial hemorrhage were higher among PWUD compared to non-PWUD.

Morever, they perfomed a sub-analysis among patients with surgical treatment, and by sex and genders.

Overall it is a very interesting paper and congrats to the authors.

I have only one minor comment:

- in figure 2: the kaplan-Meier describing the differences among the cohorts examined relative to VT/VF is indicative of statistical significance between the 2 groups, while in the text this difference is not signifcative. Please address this.

Author Response

We thank the reviewer for noticing this inconsistency. After reviewing our code, we realized that we have plotted the VT/VF subgraph using the wrong data points. This has been corrected in the revised version and now the VT/VF subgraph is concordant with the text and Table 2, i.e., the difference is not statistically significant. Please note also that the y axis scale is different in each subgraph.

Reviewer 2 Report

In this submission, the authors aimed at comparing 5-year rates of mortality and cardiovascular and cerebrovascular events after IE between people who use drugs and those who do not. A total of 7,132 PWUD patients and 7,132 propensity score-matched non-PWUD patients after a first episode of IE were studied. Matched PWUD were 41±12 years old; 52.2% men. PWUD had higher mortality vs. non-PWUD after 1 year, which was largely driven by higher mortality among female patients. PWUD also had higher rates of myocardial infarction, heart failure, ischemic stroke and intracranial hemorrhage compared to non-PWUD. Among surgically treated PWUD, interventions on the tricuspid valve were more common; however, rates of all outcomes were comparable to non-PWUD. Authors conclude by saying that PWUD had higher 5-year incidence of cardiovascular and cerebrovascular events after IE compared to non-PWUD patients. 

Some comments.

Methods. Authors must confirm for the generality of the readership if this TriNetX network is a public or a private initiative and if this is an administrative database. Both are important aspects that must be highlighted. This is regardless of HIPAA compliance. Authors must understand that a proportion of JPM readership resides out of the US. It looks as an administrative database and the community knows very well that administrative databases are of lower quality than medical databases desgined by physicians for medical purposes and not for billing only.

Why 16 years as the lower cutoff value for age? There is almost universal agreement that adults are considered as ≥18 years. Is this age 16 arbitrarily chosen? A similar problem arises with 64 as a cutoff value. An individual of 67 could easily be a PWUD regardless of other conditions. And an individual, e.g., 55 can be a PWUD with comorbidities as much older patients. Elaborate, as for the readership this sounds arbitrary, in addition as data coming from an administrative database.

Results. It looks that this network collects data mostly from the South and also from the Northeast, as in both groups around 50% of the patients came from the South. This might then bias the results towards this geographical location. 

The problem is that although authors report “all-cause mortality”, as usual, something that we know it is an statistical trick, there is no way then to make sure if mortality is actually due to addiction or not or, specifically, to infective endocarditis or not, which is the topic of the study. Furthermore, being the database administrative, it is unlikely that the actual cause of death is registered. Second, regardless of the patient numbers, it is very likely that there is simply no information about possible postmortem examinations. The reviewer is convinced that the postmortem examination rate is probably be near zero, unless authors state it otherwise.

The results about surgical treatment are not a surprise. As authors compare different populations, it is well known that the predominantly involved valve in the PWUD is the tricuspid valve and in the non-PWUD the aortic (or the mitral). This does not necessarily mean that PWUD do not have left-sided endocarditis, but the information accumulated over the past 50 years is clear with this regard in all studies. Then, no surprises.

Conclusions. Call this “Discussion” instead of “Conclusions”. Conclusion remarks must be short and strong messages to take home. Authors spend here quite a bit of time actually discussing results and other aspects of the problem. Check the instructions for authors and you will find clearly described the “Discussion” which is not the same as the “Conclusions”. This is a much better option that the “Free format” chosen.

As per the statement “…Together, these findings indicate poorer outcomes of IE in PWUD compared to IE unrelated to drug use, despite matching for other patient characteristics and medical conditions…” is rather speculative. The main issue here is to know the actual cause of death despite medical certificate. “All-cause” as stated is an unclear way to report. Then, it is possible (or not) that the worse outcomes of PWUD over time may (or may not) be related to IE. What about reinfection, suicide, killings,… which in in this supposed-to-be poorer and socially deprived populations are common. Despite AMI, AF etc. People die from many other caused than heart problems.

What is clear here is, as acknowledged by the authors, that they did not investigate the actual cause of death. Matching is nice; however, not knowing the actual cause of death renders matching in the weak statistical and philosophical side. As also agreed by authors, with the lack of robust information, everything goes in the “suggests” direction.

Identify the “limitations” and actual “conclusions” within this called “conclusions” section that must be called “discussion” section, under separate subheadings.

The limitations are rather strong and confirm the poor quality of administrative databases. This TriNetX example is a very good one. In the end and despite the large numbers and statistical approaches on a 360° perspective, authors themselves agree that their effort is not worth it due to the lack of robust information. All this is condensed in the “limitations” subsection.

Author Response

Response to Reviewer #2

In this submission, the authors aimed at comparing 5-year rates of mortality and cardiovascular and cerebrovascular events after IE between people who use drugs and those who do not. A total of 7,132 PWUD patients and 7,132 propensity score matched non-PWUD patients after a first episode of IE were studied. Matched PWUD were 41±12 years old; 52.2% men. PWUD had higher mortality vs. non-PWUD after 1 year, which was largely driven by higher mortality among female patients. PWUD also had higher rates of myocardial infarction, heart failure, ischemic stroke and intracranial hemorrhage compared to non-PWUD. Among surgically treated PWUD, interventions on the tricuspid valve were more common; however, rates of all outcomes were comparable to non-PWUD. Authors conclude by saying that PWUD had higher 5-year incidence of cardiovascular and cerebrovascular events after IE compared to non-PWUD patients. 

 We thank the reviewer for carefully evaluating our work and for providing constructive comments. 

 Methods. Authors must confirm for the generality of the readership if this TriNetX network is a public or a private initiative and if this is an administrative database. Both are important aspects that must be highlighted. This is regardless of HIPAA compliance. Authors must understand that a proportion of JPM readership resides out of the US. It looks as an administrative database and the community knows very well that administrative databases are of lower quality than medical databases designed by physicians for medical purposes and not for billing only.

 This is an important point and merits further clarification. TriNetX, LLC is a private initiative that collaborates with participating healthcare organizations (both academic and non-academic) to combine real-world clinical data with the goal to facilitate clinical trial design and conduct (by providing detailed data on patient characteristics for potential enrollment) and enable generation of real-world observational evidence, as in the case of the present study. Therefore, TriNetX is not an administrative database (and not created for billing or financial purposes), as it has been designed specifically for research purposes, and includes detailed data on demographics (within the limits of HIPAA and GDPR compliance), medical conditions, vital signs and laboratory data, procedures, and medications, using the standardized reporting systems we have described in the paper. Although TriNetX is a global research network as stated in the paper, the data from the Analytics Research Network we used for this study were derived primarily from US centers, as access to data depends on the location of the center, and we are in the US. In the revised version of the manuscript, we have clarified (Subsection 2.1, “Study Design and Data Source”):

“TriNetX, LLC, is a private initiative that partners with healthcare organizations (both academic and non-academic) to combine and harmonize real-world clinical data for research purposes (i.e., not for administrative or business purposes), with the goal of facilitating clinical trial design and conduct and enabling generation of real-world observational evidence. The harmonized data include demographics, standard measurements (including vital signs, lab results, medications, and genomic data), and diagnostic and procedural information recorded using standard coding systems, e.g., International Classification of Diseases, Tenth Revision (ICD-10) and Current Procedural Terminology (CPT) codes.”

Why 16 years as the lower cutoff value for age? There is almost universal agreement that adults are considered as ≥18 years. Is this age 16 arbitrarily chosen? A similar problem arises with 64 as a cutoff value. An individual of 67 could easily be a PWUD regardless of other conditions. And an individual, e.g., 55 can be a PWUD with comorbidities as much older patients. Elaborate, as for the readership this sounds arbitrary, in addition as data coming from an administrative database.

We fully agree with the reviewer that more clarifications are required here as otherwise the age range would seem arbitrary. The selected age range stems from the addiction and infectious disease literature (Cooper et al. Clin Infect Dis 2007, 45, 1200-1203 and authors afterward), as previous studies have shown that the accuracy of ICD codes to capture these disorders with higher sensitivity is improved when applied to this age range. Since there are no specific ICD codes for drug-use endocarditis, previous authors have found that the age range 16 to 18 had the highest combined specificity and sensitivity to identify IE specifically in PWUD from drug use and to exclude other non-drug-related causes of IE (Cooper et al. Clin Infect Dis 2007, 45, 1200-1203; Marks et al. Open Forum Infectious Diseases 2020, 7; McGrew et al. Am J Epidemiol 2021, 190, 588-599). We have made the following revisions in the Methods section (subsection 2.2 "Cohorts and Sample Selection”):

“Furthermore, in order to capture the majority of PWUD with IE while simultaneously excluding elderly patients who have higher baseline risk of IE due to age, we follow the validated methods of Cooper et al. in restricting patient age to 16-64. This methodology has been found to maximize specificity, despite age cutoffs which do not typically correspond to adult age cutoffs reported in the literature.”

Results. It looks that this network collects data mostly from the South and also from the Northeast, as in both groups around 50% of the patients came from the South. This might then bias the results towards this geographical location. 

 We thank the reviewer for this comment and agree that mention of geographic limitations should be acknowledged formally. We have made the following note in the Limitations section (please note that we also outlined our Discussion per the reviewer’s suggestion):

“Sixth, the majority of data in both cohorts represented the South and Northeast regions of the United States. Although the geographic distribution was comparable between cohorts, limited availability of data from other regions may potentially bias results based on location, which we acknowledge and furthermore suggest for future topic of investigation.”

The problem is that although authors report “all-cause mortality”, as usual, something that we know it is a statistical trick, there is no way then to make sure if mortality is actually due to addiction or not or, specifically, to infective endocarditis or not, which is the topic of the study. Furthermore, being the database administrative, it is unlikely that the actual cause of death is registered. Second, regardless of the patient numbers, it is very likely that there is simply no information about possible postmortem examinations. The reviewer is convinced that the postmortem examination rate is probably be near zero, unless authors state it otherwise.

 We completely agree with the reviewer that besides general assessment of mortality, a causal association between sequelae of infective endocarditis (and of potentially continuing drug use among drug users) and cause of death is impossible to ascertain from these data. We have therefore added the following part to the Limitations section:

“Fifth, we acknowledge that our primary endpoint of all-cause mortality may not represent death directly as a result of IE or sequalae of IE. As postmortem data are unavailable and linkage to death certificates would not be possible due to the deidentified nature of the TriNetX data, exact causes of death are impossible to ascertain; nevertheless, discrepancies in all-cause mortality after IE in PWUD still represents an important metric to investigate and follow.”

The results about surgical treatment are not a surprise. As authors compare different populations, it is well known that the predominantly involved valve in the PWUD is the tricuspid valve and in the non-PWUD the aortic (or the mitral). This does not necessarily mean that PWUD do not have left-sided endocarditis, but the information accumulated over the past 50 years is clear with this regard in all studies. Then, no surprises.

 We agree with the reviewer’s comment and with the fact that right-sided endocarditis is more common among PWUD. In this study, the PWUD and non-PWUD subsets were matched for several key variables, although not for valves specifically. Furthermore, our investigation is in cardiovascular outcomes after infective endocarditis events, so we believe it is necessary to include a discussion about the general trend as well as exceptions with IE location given that the assumption of right-sided infective endocarditis without clinical consideration of left-sided involvement has been shown to influence outcomes for PWUD who are already at risk for poorer quality treatment. (Clarelin et al Sci Rep. 2021;11(1):1177; Seghatol and Grinberg Echocardiography. 2002;19(6):509-511)

Conclusions. Call this “Discussion” instead of “Conclusions”. Conclusion remarks must be short and strong messages to take home. Authors spend here quite a bit of time actually discussing results and other aspects of the problem. Check the instructions for authors and you will find clearly described the “Discussion” which is not the same as the “Conclusions”. This is a much better option that the “Free format” chosen.

We thank the reviewer for alerting us to this. We fully agree and have changed our “Conclusions” section to separate “Discussion,” “Limitations,” and “Conclusions” sections and believe this has improved the quality of our manuscript.

As per the statement “…Together, these findings indicate poorer outcomes of IE in PWUD compared to IE unrelated to drug use, despite matching for other patient characteristics and medical conditions…” is rather speculative. The main issue here is to know the actual cause of death despite medical certificate. “All-cause” as stated is an unclear way to report. Then, it is possible (or not) that the worse outcomes of PWUD over time may (or may not) be related to IE. What about reinfection, suicide, killings,… which in in this supposed-to-be poorer and socially deprived populations are common. Despite AMI, AF etc. People die from many other caused than heart problems.

 We thank the reviewer for this comment. Regarding the limitations of using all-cause mortality as a metric, we agree with the reviewer and have provided a response in a previous comment. Nevertheless, in this study we emphasize several cardiovascular and cerebrovascular outcomes other than mortality, all of which are important metrics to consider in the long-term health of PWUD after infective endocarditis episodes. In addition, as the addiction medicine and social science literature has pointed out, mortality among PWUD is not as common from violence as previously thought (Bech et al. BMC Health Serv Res. 2019;19(1):440; Santo et al. JAMA Psychiatry. 2021 Sep 1;78(9):1044). Furthermore, especially in the recent and current landscape of the opioid crisis in the United States, mortality from medical sequalae of drug use is more common than homicide or suicide. Thus, we believe that reporting all-cause mortality, with all the limitations that we have acknowledge in the revised limitations subsection, as well as other cardiovascular outcomes of infective endocarditis in PWUD are important. 

What is clear here is, as acknowledged by the authors, that they did not investigate the actual cause of death. Matching is nice; however, not knowing the actual cause of death renders matching in the weak statistical and philosophical side. As also agreed by authors, with the lack of robust information, everything goes in the “suggests” direction.

 In agreement with our responses above, we agree that all-cause mortality has technical limitations when used in large data research, as individual adjudication of cause of death would be impossible at this scale. We emphasize however that our findings are not limited to mortality, but to other cardiovascular outcomes which differ based on drug use history. Unfortunately, exact causes of death from a death certificate cannot be ascertained from the TriNetX database, but the combination of data from other outcomes as well as general trends noted in addiction medicine, cardiology, and infectious disease research points to clear discrepancies between PWUD and non-PWUD, which our data corroborates. Please refer to our previous comment regarding the cause-of-death limitations with the TriNetX data/

Identify the “limitations” and actual “conclusions” within this called “conclusions” section that must be called “discussion” section, under separate subheadings.

We have revised the structure accordingly as stated in a previous comment.

The limitations are rather strong and confirm the poor quality of administrative databases. This TriNetX example is a very good one. In the end and despite the large numbers and statistical approaches on a 360° perspective, authors themselves agree that their effort is not worth it due to the lack of robust information. All this is condensed in the “limitations” subsection.

 We acknowledge the limitations of our paper, many of which are limitations of large healthcare database research in general and are not specific to TriNetX. Also, we would like to re-state that TriNetX is a not an administrative database and the depth of the data surpasses that of administrative databases allowing for detailed matching of clinical characteristics. Nevertheless, we strive to compare our cohorts using methodology previously used and validated across several fields of research including using high sensitivity and specificity ICD criteria and propensity score matching to maximize reliability of our findings. Furthermore, we specify our limitations throughout the manuscript and as the reviewer comments, are transparent with the fact that our conclusions are suggestive rather than causative. We use this language throughout our manuscript in order to provide readers with an accurate reflection of our findings.

Additional References (In response letter):

Seghatol F, Grinberg I. Left-sided endocarditis in intravenous drug users: a case report and review of the literature. Echocardiography. 2002;19(6):509-511

Clarelin A, Rasmussen M, Olaison L, Ragnarsson S. Comparing right- and left sided injection-drug related infective endocarditis. Sci Rep. 2021;11(1):1177.

Bech AB, Clausen T, Waal H, ŠaltytÄ— Benth J, Skeie I. Mortality and causes of death among patients with opioid use disorder receiving opioid agonist treatment: a national register study. BMC Health Serv Res. 2019;19(1):440

Santo T Jr, Clark B, Hickman M, et al. Association of Opioid Agonist Treatment With All-Cause Mortality and Specific Causes of Death Among People With Opioid Dependence: A Systematic Review and Meta-analysis. JAMA Psychiatry. 2021 Sep 1;78(9):1044

Reviewer 3 Report

The study “Five-Year Cardiovascular Outcomes after Infective Endocarditis 2 in Patients with Versus without Drug Use History” is important in presenting the longer-term outcomes among infective endocarditis (IE) patients. Use of TriNetX data is efficient and the authors are commended for investigating several outcomes.

 1.       The inclusion-exclusion criteria needs to be clearer.

2.       Lines 115-116: “We compared mortality and major cardiovascular and cerebrovascular morbidity at 5 years after an initial IE episode among PWUD and non-PWUD.” I want to confirm that these patients had not had a previous episode of IE. Was this their first episode? Were the patients with a previous episode excluded from cohort selection?

3.       Lines 119-121: “For each morbidity endpoint, we excluded patients with prior history of the endpoint of interest (i.e., before the index IE episode) from our primary and secondary analyses.” But were these patients included in the cohort and later excluded from analyses? Including them in the cohort can cause a bias.

4.       Table 1 shows Standardized Mean Difference (SMD). For comparing the PWUD and Non-PWUD cohorts before Matching, the norm in our field is to use chi-squared test and present the p-values for categorical variables. Since it is also difficult to interpret SMD for categorical variables in the unmatched groups, I would suggest the authors present the chi-squared p-values.

5.       It is interesting to note that in Table 1 the proportion of patients with almost every comorbidity was higher among the Non-PWUD after matching. However, the reverse was true before matching.

6.       Table 1 shows patients were matched on comorbidities, such as Atrial fibrillation, Heart Failure, and cerebrovascular diseases. Did the patients have these comorbidities prior to the IE diagnosis?

7.       Some of the outcomes are the same as comorbidities. If the comorbidities occurred prior to IE diagnosis, they cannot be outcomes. For example, Atrial fibrillation and Heart Failure. Patients should not have been matched on these outcomes.

8.       Since this is a retrospective cohort study, patients cannot be matched on outcomes. A hazard ratio should not be calculated for those matched outcomes (Atrial fibrillation, Heart Failure).

9.       In addition, Stroke and Intracranial Hemorrhage were outcomes in Table 2 but already matched on cerebrovascular diseases in Table 1. Similarly, Myocardial Infarction and VT/VF were outcomes in Table 2, but already matched for several of the cardiovascular diseases in Table 1. Overmatching can result in biased estimates.

10.   Once the matching criteria are revised per the above suggestions, the estimates are likely to change. Therefore, at this time commenting on those estimates might not be useful.

11.   There are a few more recent publications that authors can use as reference to strengthen their paper.

Author Response

Response to Reviewer #3

1.     The inclusion-exclusion criteria needs to be clearer.

We thank the reviewer for this comment and agree that inclusion criteria should be better clarified. Methods section 2.2 now reads:

“ We queried the TriNetX Research Network database from January 1, 2010-December 31, 2020, for patients with first-episode infective endocarditis (IE) with and without history of drug use according to the methodology used in a previous pivotal retrospective cohort study [7, 8]]. Inclusion criteria for IE were (1) ICD-10 codes I33, I38, and I39; (2) age 16 to 64 at the time of the index IE event; (3) for PWUD designation, ICD-10 codes: F11, F13, F14, F15, F16, F18, F19, T40, and O99.32. Patients with a previous episode of IE were excluded.

To capture the majority of PWUD with IE while simultaneously excluding elderly patients who have higher baseline risk of IE due to age, we followed the validated methods of Cooper et al. in restricting patient age to 16-64 [11]. This methodology has been found to maximize specificity, despite age cutoffs which do not typically correspond to adult age cutoffs reported in the literature [11]. In the ICD-10 coding system, there are no specific codes for IE related to drug use, nor are there specific codes for injection drug use. Therefore, we drew on published and validated methods to create a sensitive list of drug use codes which were used to capture PWUD: [7,11-13]. Importantly, there are no specific codes to distinguish IE by common micro-organisms (i.e., separate codes for staphylococcal IE versus streptococcal IE, etc.) as these are contained in the ICD-10 codes I33, I38, and 139 [9,10]. We thus obtained two primary cohorts of interest: PWUD with IE and non-PWUD with IE.”

  1. Lines 115-116: “We compared mortality and major cardiovascular and cerebrovascular morbidity at 5 years after an initial IE episode among PWUD and non-PWUD.” I want to confirm that these patients had not had a previous episode of IE. Was this their first episode? Were the patients with a previous episode excluded from cohort selection?

We thank the reviewer for this clarifying question. The reviewer is correct in that we only queried the TriNetX database for first-episode IE. Consequently, we have clarified this in the manuscript with supporting literature for this methodology: pg 2 line 93-96

“We queried the TriNetX Research Network database from January 1, 2010-December 31, 2020, for patients with first-episode infective endocarditis (IE) with and without history of drug use according to the methodology used in a previous pivotal retrospective cohort study [7,8].”

  1. Lines 119-121: “For each morbidity endpoint, we excluded patients with prior history of the endpoint of interest (i.e., before the index IE episode) from our primary and secondary analyses.” But were these patients included in the cohort and later excluded from analyses? Including them in the cohort can cause a bias.

Excluding upfront all patients with any prevalent cardiovascular event of interest (i.e., history of myocardial infarction, heart failure, atrial fibrillation, cardiac arrest, or Ischemic/hemorrhagic stroke) would effectively exclude 50% of both cohorts (Table 2), introducing significant bias and limiting the generalizability of our findings. Instead, we only excluded patients with a history of the event of interest, e.g., atrial fibrillation, when evaluating incident events and comparing risks. Removing patients with e.g., baseline atrial fibrillation, from matched cohorts does not introduce bias, as the proportion of removed patients is by definition similar between the matched cohorts.

From a conceptual perspective, removing patients with any cardiovascular condition of interest at baseline would substantially underestimate risk for incident events. For example, the risk for future heart failure would be substantially lower vs. actual risk if we were to remove patients with previous myocardial infarction or atrial fibrillation at baseline.

We have clarified this further in the manuscript, pg 3 lines 120-123:

“For each morbidity endpoint, we excluded patients with prior history of the endpoint of interest (i.e., before the index IE episode) from our primary and secondary analyses, thereby assessing only for new-onset or newly diagnosed conditions of interest. Of note, removing patients with a prevalent event/condition of interest from matched cohorts does not introduce bias, as the proportion of removed patients is by definition similar between the matched cohorts.”

  1. Table 1 shows Standardized Mean Difference (SMD). For comparing the PWUD and Non-PWUD cohorts before Matching, the norm in our field is to use chi-squared test and present the p-values for categorical variables. Since it is also difficult to interpret SMD for categorical variables in the unmatched groups, I would suggest the authors present the chi-squared p-values.

Most authorities in propensity score matching recommend using the SMD over P values (based on e.g., chi-square or t tests) for assessment of cohort balance. First, the P values need to be associated with some hypothesis testing to be meaningful, which is not the case when matching two subsets of parent cohorts, i.e., there is no underlying hypothesis. Second, unlike the P value, the SMD is not dependent on sample size. For example, a small sample size can lead to non-significant P values when comparing a characteristic between cohorts even if the difference in the distribution is clinically meaningful, therefore giving the wrong impression of a well-balanced characteristic. Conversely, when comparing large sample sizes, as in our case, most P values are significant even for negligible differences and despite matching with a very tight caliper (0.1 of standard deviation). For example, Austin explicitly mentions that statistical hypothesis testing is not appropriate and recommends the SMD (Austin PC, Circ Cardiovasc Qual Outcomes 2008). In the revised manuscript, we offer a thorough documentation for the use of SMD, and we describe the definition to orient readers:

“We used the standardized mean difference (SMD) as the preferred method to assess balance between cohorts, as statistical hypothesis testing methods are not appropriate when balancing cohorts with propensity score matching. The SMD is calculated as the difference in the means divided by the estimate of the within-group standard deviation [13].”

  1. It is interesting to note that in Table 1 the proportion of patients with almost every comorbidity was higher among the Non-PWUD after matching. However, the reverse was true before matching.

This is to be expected, as we restricted the age to 16-64 before cohort inception, for the reasons explained in the manuscript (i.e., increased specificity of drug-associated infective endocarditis), leading to only a small age differential between the unmatched cohorts. In the same age group, it has been consistently reported that PWUD have more concomitant medical conditions and neglected health issues compared to non-PWUD (Barry et al., Psych. Serv. 2014; 65(10): 1269-72; Biancarelli et al., Drug & Alc. Dep. 2019; 198:80-6; Cullen et al., BMC Fam. Pract. 2009; 10:25). The nominally higher percentage of several comorbidities in the PWUD group after matching is probably due to chance – and in all instances the difference is less than 10% (SMD <0.1).     

  1. Table 1 shows patients were matched on comorbidities, such as Atrial fibrillation, Heart Failure, and cerebrovascular diseases. Did the patients have these comorbidities prior to the IE diagnosis? 

Yes, these were prevalent diagnoses, i.e., have been recorded before the index health care encounter. Please see further explanation in response #2 above and #7 below. We have clarified this in the revised version of the manuscript.

  1. Some of the outcomes are the same as comorbidities. If the comorbidities occurred prior to IE diagnosis, they cannot be outcomes. For example, Atrial fibrillation and Heart Failure. Patients should not have been matched on these outcomes. 

The reviewer is right that we should have clarified this point better. For each outcome, patients with the corresponding condition at baseline have been excluded from the analysis. Although this appears under 2.3. Outcomes in the original version of the manuscript, probably many readers expect to see this in the Statistical Analysis section. We have added this information in 2.4. Statistical Analysis: pg 3, lines 141-143:

“Of note, patients with the event or condition of interest present at baseline were excluded in secondary analyses for the corresponding outcome, to query only for new-onset events as described above. For example, patients with baseline atrial fibrillation were excluded from incident atrial fibrillation analysis.”

To avoid any ambiguity, we have also included it in the Table 2 footnote and Figure 2 legend:

Of note, we excluded patients who had recorded history of each outcome of interest at baseline, respectively (i.e. those with known AF were excluded from AF analysis) in order to only capture new-onset events/conditions.”

  1. Since this is a retrospective cohort study, patients cannot be matched on outcomes. A hazard ratio should not be calculated for those matched outcomes (Atrial fibrillation, Heart Failure).

We matched the patients for baseline characteristics, not outcomes. Five-year outcomes were compared between matched cohort after excluding those who have already experienced the event of interest (e.g., atrial fibrillation) at baseline. We hope that the previous clarification resolved this potential miscommunication.

  1. In addition, Stroke and Intracranial Hemorrhage were outcomes in Table 2 but already matched on cerebrovascular diseases in Table 1. Similarly, Myocardial Infarction and VT/VF were outcomes in Table 2, but already matched for several of the cardiovascular diseases in Table 1. Overmatching can result in biased estimates.

It is necessary to include prevalent conditions strongly associated with the outcome of interest. For example, although we excluded patients with prevalent myocardial infarction (i.e., history of myocardial infarction at baseline) when evaluating myocardial infarction as an outcome (that is, incident myocardial infarction), it is necessary to include the presence of ischemic heart disease at baseline as a matching characteristic for a valid comparison, because patients with existing ischemic heart disease (even without a history of myocardial infarction) are at higher risk of experiencing myocardial infarction during follow up.

  1. Once the matching criteria are revised per the above suggestions, the estimates are likely to change. Therefore, at this time commenting on those estimates might not be useful.

We hope that the previous clarifications resolved any miscommunication on matching and related issues. Based on the above, we believe that our estimates are valid within the limitations of the nature of the database.

  1. There are a few more recent publications that authors can use as reference to strengthen their paper.

We have added the following references:

  1. Bhandari R, Alexander T, Annie FH, et al. Steep rise in drug use-associated infective endocarditis in West Virginia: Characteristics and healthcare utilization. PLoS One. 2022;17(7):e0271510.
  2. Yucel E, Bearnot B, Paras ML, et al. Diagnosis and Management of Infective Endocarditis in People Who Inject Drugs: JACC State-of-the-Art Review. J Am Coll Cardiol. 2022;79(20):2037-2057. 
  3. Austin PC. Primer on statistical interpretation or methods report card on propensity-score matching in the cardiology literature from 2004 to 2006: a systematic review.Circ Cardiovasc Qual Outcomes. 2008;1(1):62-67. 

Round 2

Reviewer 2 Report

The authors come back with a fully revised version of their original submission. Authors address important issues like the actual nature of the database/TriNetX network, which seems to be private initiative.

Authors choose specific age cutoff values and they justify this based on a specific isolated article on infectious disease. This continues to be rather poor despite the topic.

Authors acknowledge the many issues representing strong limitations like the geographical distribution and “all-cause” mortality. This is a very serious issue that they acknowledge in full.

Authors have gone through formalities.

The major problem of not investigating the cause of death cannot be counteracted with design modifications.

Author Response

We thank the reviewer for the suggestions provided previously. At this time, no new suggestions are actionable. 

Reviewer 3 Report

The authors have answered my questions and revised the manuscript per my suggestions from the first round of review.